# Continuous Goal Sampling: A Simple Technique to Accelerate Automatic Curriculum Learning

## Abstract

Goal-conditioned reinforcement learning (RL) tackles the problem of training an RL agent to reach multiple goals in an environment, often with sparse rewards only administered upon reaching the goal. In this regard, automatic curriculum learning can improve an agent's learning by sampling goals in a structured order catered to the agent's current ability. This work presents two contributions to improve learning in goal-conditioned RL environments. First, we present a simple, algorithm-agnostic technique to accelerate learning by continuous goal sampling, in which an agent's goals are sampled and changed multiple times within a single episode. Such continuous goal sampling enables faster exploration of the goal space and allows curriculum methods to have a more significant impact on an agent's learning. Second, we propose VDIFF, an automatic curriculum learning method that uses an agent's value function to create a self-paced curriculum by sampling goals on which the agent is demonstrating high learning progress. Through results on 17 multi-goal robotic environments and navigation tasks, we show that continuous goal sampling, combined with VDIFF or existing curriculum learning methods, results in performance gains over state-of-the-art methods.

## 1 Introduction

Recent successes in deep reinforcement learning (DRL) have proven that it can tackle complex sequential decision-making problems for tasks in diverse domains such as robotics, video games, and traffic control (Andrychowicz et al., 2020; Vinyals et al., 2019; Li et al., 2016). Building upon the success of DRL in solving specific robotic tasks, the next step is the development of more general-purpose methods that can solve multiple tasks. With this objective in mind, the area of goal-conditioned RL (GCRL) (Schaul et al., 2015) has received increased attention as an extension of standard RL. Here, agents are trained to learn a policy that can achieve multiple goals, with sparse rewards being provided when the agent achieves the desired goal (Andrychowicz et al., 2017).

A natural question arises from the GCRL formulation – given a set or a distribution of goals, in what order should they be presented to the learning agent? A naive strategy is to sample these goals uniformly from the goal distribution. This strategy has two drawbacks. First, humans and other biological agents do not have a random, unstructured order to their learning (Ferster & Skinner, 1957). For example, a baby does not start walking before it can learn to crawl. Using the same intuition, at the start of learning, when an RL agent essentially has a random policy, sampling hard-to-achieve goals will result in no learning signal and wasted samples with high probability. Second, given that the training time is limited, it is often not even possible to adequately cover the entire goal space for high-dimensional problems by random sampling. To address these challenges, curriculum learning methods structure an agent's learning by organizing the order in which goals or tasks are presented to the agent (Soviany et al., 2022). The fundamental intuition behind many of these methods is to sample goals that are neither too easy nor too hard and are thus maximally informative to the agent. Although driven by the same intuition, the technical objectives used in prior work vary significantly. For example, Matiisen et al. (2019) and Portelas et al. (2020) propose to sample tasks on which an agent has high learning progress (LP), while VDS (Zhang et al., 2020) proposes to sample goals that have high expected epistemic uncertainty.

While curriculum learning methods have led to significant improvement over vanilla GCRL, they are still quite sample inefficient in sparse reward settings, especially for problems with high dimensional goal spaces. One of the reasons behind this is that the standard GCRL setup constrains an agent to have a single goal for the entire length of an episode. As a result, episodes where the goal is too hard or too easy to achieve are non-informative and waste agent interactions. The single-goal framework also restricts curriculum learning methods to sampling goals from the initial state and does not give them the flexibility to change an agent's goal online based on its current state. However, since goals do not affect transition dynamics, this constraint is not required to be enforced but is simply a result of the chosen conventional framework. Using this observation and the recent success of curriculum learning methods, this work presents two contributions to accelerate and improve GCRL.

First and as the main focus of this work, we present Continuous Goal sampling, a simple technique that proposes a novel extension to GCRL which can accelerate a wide range of curriculum learning algorithms. In continuous goal sampling, goals are sampled multiple times in an episode, instead of the standard practice of sampling a goal only at the start of an episode. Second, inspired by the success of recent LP-based curriculum methods in multi-task RL (Matiisen et al., 2019; Portelas et al., 2020), we reformulate the LP objective to enable it to work in sparse reward settings with random initial states. We achieve this by using a current and a lagged value function to approximate the expected LP of a goal. Since value functions are a part of most current deep RL algorithms, we reuse them off the shelf for LP computation. Our proposed automatic curriculum learning algorithm, referred to as VDIFF, requires no additional learning, has little to no computational overhead, and is agnostic to the base RL algorithm.

We present results on a set of 14 benchmark manipulation environments (Plappert et al., 2018; Gallouédec et al., 2021) and 3 maze navigation environments (Zhang et al., 2020). Through these results, we show that resampling improves the sample efficiency and performance of explicit curriculum learning algorithms (VDS and VDIFF), implicit curriculum learning methods (HER), and vanilla RL algorithms in multiple environments. Finally, we also show that our proposed curriculum learning method VDIFF is able to outperform existing explicit curriculum learning methods in GCRL. Anonymized code is available here.

## 2 BACKGROUND

### 2.1 GOAL-CONDITIONED RL

Goal-conditioned RL (GCRL) (Schaul et al., 2015) aims to train an agent to achieve multiple goals in an environment. Formally, it can be described as a finite-horizon Markov decision process (MDP) defined by the tuple $(S, G, A, R, T, \rho, \gamma)$, where $S$ is the state space, $A$ is the action space, $R : S \times A \times G \to \mathbb{R}$ is the the reward function, $T : S \times A \times S \to [0, 1]$ is the state transition function, $\rho(s_0)$ is the initial state distribution, and $\gamma \in [0, 1]$ is the *discount factor*. At the start of each episode, a goal $g$ is sampled from $G$ and the objective is to learn a policy $\pi(a_t|s_t, g)$ which maximizes the expected value $V$, which is defined as the sum of discounted rewards $V(s_0, g) = \mathbb{E}[\sum_{t=0}^{T} \gamma^t R(s_t, a_t, g)]$. Given the dependence of $R$ on $g$, GCRL can also be viewed as the problem of learning a policy $\pi$ over a distribution of reward functions $R^g$ parameterized by a goal $g$.

The inherent binary structure of GCRL setup allows for the definition of a sparse indicator reward function that indicates if an agent has achieved the given goal $g$ (Andrychowicz et al., 2017). An agent receives a reward of 0 and is said to have achieved goal $g$ when $d(s_t, g) < \epsilon$, where $d(.,.)$ is some distance function in goal space and $\epsilon$ is the acceptance threshold. In all other cases, the agent receives a reward of $-1$. In this work, an episode terminates as soon as an agent achieves the goal.

### 2.2 CURRICULUM LEARNING SETUP

In goal-conditioned RL, a goal $g$ is sampled from $G$ according to some probability distribution $p(g|s_0)$, where $s_0$ is the initial state. The task of a curriculum learning method is to design $p(g|s_0)$ to enable the sampling of meaningful goals that can both improve and accelerate an agent's learning. Curriculum learning methods dynamically change $p(g|s_0)$ as an agent's policy evolves during training. To make this dependence on policy more explicit, we will denote the goal distribution designed by a curriculum method as $p^\pi(g|s_0)$. Existing curriculum methods generally construct

---

**Algorithm 1** Continuous Goal Sampling Algorithm

---

**Input:** Max Episode Length $T_{max}$, Resample Frequency $R$, policy $\pi$
**Initialize:** Initial state $s_0 \sim \rho(s_0)$, Initial goal $g_{current} \sim p^\pi(g|s_0)$ [Using equation 6]
**for** $t = 0, 1, ...T_{max}$ **do**
    $a_t \sim \pi(s_t, g_{current})$
    $s_{t+1}, r_{t+1}, done = environment\_step(a_t)$
    **if** $done$ **then**
        $break$
    **end if**
    **if** $t$ **mod** $R == 0$ **then**
        $g' \sim p^\pi(g'|s_{t+1})$                           $\triangleright$ In practice, done using Equation 6
        $g_{current} = g'$
    **end if**
**end for**

---

$p^\pi(g|s_0) \propto f(s_0, g)$, where $f$ is some function encoding the curriculum objective. For example, in VDS (Zhang et al., 2020), $f$ is the epistemic uncertainty of sampling goal $g$ from initial state $s_0$. That is, the objective of VDS is to sample goals which have high epistemic uncertainty. Similarly, in ALP-GMM and R-IAC (Portelas et al., 2020; Baranes & Oudeyer, 2009), $f$ is the learning progress (LP). For vanilla goal-conditioned RL (Section 2.1) in which goals are sampled randomly, $f$ reduces to an uniform distribution $U(g)$ over the goal space. Finally, note the dependence of $p^\pi(g|s_0)$ on the initial state $s_0$. While it is possible to drop $s_0$ if the environment always resets to a fixed state, such as in GoalGAN (Florensa et al., 2018), that is not the case in many real-world settings.

## 3 METHOD

In this section, we first introduce continuous goal sampling, a simple technique to accelerate GCRL by sampling goals multiple times in an episode. Then, we introduce VDIFF, a new automatic curriculum method for GCRL, which utilizes value functions to compute learning progress (LP).

### 3.1 CONTINUOUS GOAL SAMPLING

In continuous goal sampling, in addition to setting a new goal at the start of an episode (standard practice), we continuously sample and set a new goal every $R$ timesteps in an episode, where $R$ is less than the maximum episode length. The main idea behind this is that the standard GCRL framework, which has enjoyed notable success in robotic tasks (Andrychowicz et al., 2017), can be even further extended by removing the constraint of restricting an episode to a single goal. The removal of this constraint is made possible by observing that a goal $g$ is a parameterization of the reward function and has no effect on environment transition dynamics. We incorporate resampling into our training framework by sampling a new goal $g'$ every $R$ timesteps during an episode, where $R$ is referred to as the resample frequency. The probability of sampling a specific goal $g$ is given by $p^\pi(g|s_t)$, where $p^\pi(g|s_t) \propto f(s_t, g)$, as described in Section 2.2. The continuous goal sampling algorithm is formally described in Algorithm 1.

### 3.2 LEARNING PROGRESS BASED CURRICULUM

Existing curriculum methods using learning progress (LP) as an objective have shown great promise in the domain of multi-task RL with dense rewards and low-dimensional tasks (Portelas et al., 2020; Romac et al., 2021). However, they are not directly applicable to goal-conditioned RL, which not only has binary episode returns, but also has random initial states that are possibly high dimensional. In this section, drawing on the success of LP in multi-task RL, we present VDIFF, an LP-based automatic curriculum learning method for GCRL.

---

**Algorithm 2** LP-based Curriculum Learning Algorithm

---

**Input:** Initial Parameters $(\pi, V, V^T, V^{T'}) = (\phi_0, \theta_0, \theta'_0, \theta'')$, Blackbox Learning Algorithm A,
       Total Training Episodes $N_{max}$, Fit Rate $F$
**Output:** Learned policy parameters $\phi_N$
**Initialize:** $\theta'_0 = \theta_0$, $\theta'' = \theta_0$
**for** $i = 0, 1, ... N_{max}$ **do**
    Sample initial state $s_0 \sim \rho(s_0)$
    Sample set of $N$ random goals $\mathbb{G}$ using $U(g)$
    **for** $g \in \mathbb{G}$ **do**
        $LP(s_0, g) = V^T(s_0, g; \theta'_i) - V^{T'}(s_0, g; \theta'')$
    **end for**
    Sample $g \sim p(g|s_0)$ [Using equation 6]
    Collect trajectory $\tau_i(\pi_{\phi_i}|g)$
    Update Parameters:
        $\phi_{i+1} \leftarrow Update\_Policy(A, \phi_i, \theta_i, \theta'_i)$
        $\theta_{i+1} \leftarrow Update\_Value(A, \phi_i, \theta_i, \theta'_i)$
        $\theta'_{i+1} \leftarrow Update\_Target\_Value(\theta_{i+1}, \theta'_i)$ [Using equation 3. Generally inbuilt in A]
    **if** $i$ **mod** $F == 0$ **then**
        $\theta'' \leftarrow \theta'_{i+1}$
    **end if**
**end for**
**Return:** $\phi_N$

---

### 3.2.1 CURRICULUM DESIGN

We define the Learning Progress (LP) of a state and goal tuple $(s_t, g)$ as the change in the expected discounted returns of reaching the goal from that state after some specified $\Delta T$ training episodes.

$$LP(s_t, g) = V(s_t, g) - V'(s_t, g) \tag{1}$$

where $V$ is the current value function and $V'$ is the old value function, i.e., the value function $\Delta T$ training episodes ago.

Goals exhibiting high LP under this formulation are the ones in which the agent is progressing the fastest and should be sampled more aggressively. To formalize this notion, we propose to sample goals proportional to their learning progress, i.e. $p^\pi(g|s_t) \propto LP(s_t, g)$. In addition to allowing self-paced learning for the agent, using value functions to construct a curriculum has little computational overhead since the value functions learned by base RL algorithms can be reused off the shelf.

One downside of using value functions for LP inference is that they can be noisy because i) they are trained over data collected by an agent which is constantly exploring, and ii) because, at any iteration, they are trained using a small, random batch of data that could be biased. To address this, we use Polyak averaging (Polyak & Juditsky, 1992) to obtain smoothed value functions for LP inference. Smoothed value functions are a part of many deep RL algorithms and are referred to as 'target' functions because they are used to obtain the target for temporal difference (TD) updates. LP is now computed as -

$$LP(s_t, g) = V^T(s_t, g) - V^{T'}(s_t, g) \tag{2}$$

where $V^T$ is the target value function and $V^{T'}$ is the target value function $\Delta T$ training episodes ago. At each training step, the parameters $\theta'$ of $V^T$ are updated using -

$$\theta' = \alpha\theta' + (1 - \alpha)\theta \tag{3}$$

where $\theta$ are the parameters of the value function $V$, $\theta'$ are the parameters of the target value function $V^T$, and $\alpha$ is the Polyak coefficient.

The smoothed target value functions provide a cleaner, less noisy LP signal. This does not come for free, as the smoothed update causes $V^T$ to lag behind $V$. However, in practice, the gains from the reduced noise far outweigh the slowdown from the lag. This is formally defined in Algorithm 2.

**Selective Resampling:** In some situations, the current goal $g$ might be optimally challenging for an agent and sampling a new goal $g'$ will not add value. More formally, it makes intuitive sense to

sample a new goal only if the expected gain of sampling a new goal is greater than the estimated gain of the current goal. Thus, we sample a new goal if -

$$\mathbb{E}_{g' \sim p(g|s_t)}[f(s_t, g')] > f(s_t, g) \tag{4}$$

We use selective resampling where the expected value of sampling a new goal is computed as -

$$
\begin{aligned}
\mathbb{E}_{\mathbb{G}}[LP(s_t, g)] &= \sum_{i=1}^{N} [p(s_t, g^{(i)}) \times LP(s_t, g^{(i)})] \\
&= \frac{1}{Z} \sum_{i=1}^{N} [LP(s_t, g^{(i)}) \times LP(s_t, g^{(i)})] \quad \text{[Using Eqn 6]}
\end{aligned}
\tag{5}
$$

where $\mathbb{G}$ is a set of $N$ random goals and $Z$ is a normalization constant. Selective resampling is formally described in Algorithm 3.

Our final algorithm, referred to as VDIFF, develops an adaptive, self-paced curriculum for an agent without any extra learning-based component or encoded domain knowledge, thus allowing for easy integration into most existing GCRL frameworks.

## 4 EXPERIMENTAL RESULTS

In this section, we first detail the experimental setup by describing the environments tested, training algorithm and the baseline methods. Next, through a series of experiments, we analyze the performance of continuous goal sampling and VDIFF on a diverse set of multi-goal environments.

### 4.1 EXPERIMENTAL SETUP

**Environments:** We run experiments on 14 manipulation goal-conditioned tasks and 3 maze-navigation environments. These include 12 environments of the OpenAI Fetch and Hand environments (Plappert et al., 2018) which serve as benchmarks for goal-based RL and 2 tasks (Flipping and Stacking) from the panda-gym environments (Gallouédec et al., 2021). The 3 maze-navigation environments are adapted from Zhang et al. (2020). All environments use the sparse reward function detailed in Section 2.1.

**Training details:** We use Soft-Actor Critic (SAC) (Haarnoja et al., 2018) as the base RL algorithm for our method. All baseline methods also use SAC with identical hyperparameter settings to enable a fair comparison. VDIFF uses, without alteration, the target value function $V^T(s, g; \theta')$ of SAC. Additionally, to compute $V^{T'}(s, g)$, we store additional parameters $\theta''$ which are synced periodically with $\theta'$ as described in Algorithm 2. We also augment SAC by incorporating Hindsight Experience Replay (HER) (Andrychowicz et al., 2017), a highly effective relabeling technique that uses future states reached in a trajectory as virtual goals to obtain additional learning signal. However, since HER is constrained to work only for off-policy algorithms in sparse reward domains, we also present results of SAC with no HER. We train 2 variants of VDIFF:

- **VDIFF:** Uses the LP-based curriculum described in Section 3.2.1, but does NOT use continuous goal sampling.
- **VDIFF-R:** Uses both the LP-based curriculum and continuous goal sampling.

**Baseline methods:** We compare our method with the following baseline methods:

- **Random:** Random is a simple baseline method in which goals are sampled randomly from the goal space.
- **HER:** HER is a relabeling technique which augments the buffer of base RL algorithms with fictitious data to obtain additional learning signal. Since HER is not an explicit curriculum method, it can be used alongside other explicit curriculum methods. In vanilla HER, goals are sampled randomly using a uniform distribution $U(g)$ i.e., HER has no explicit curriculum.

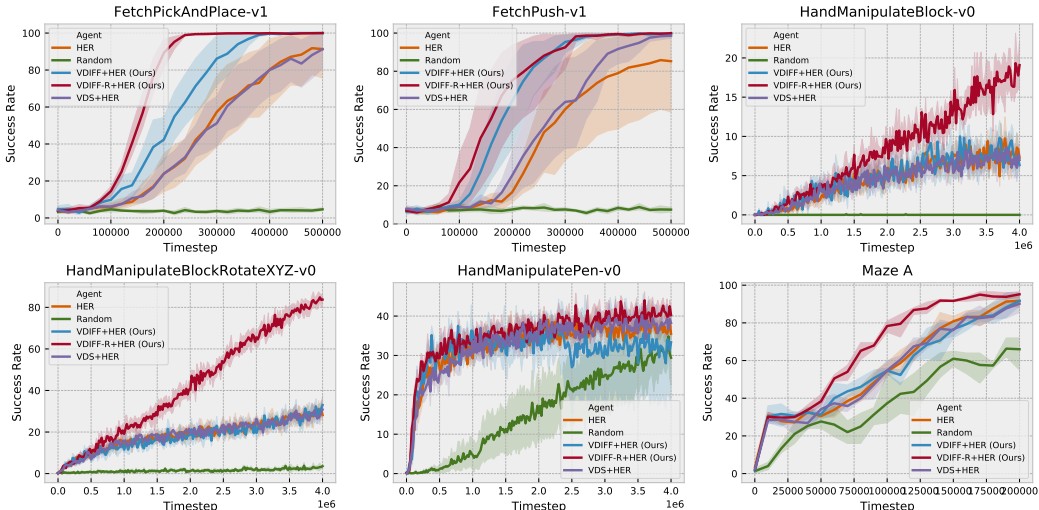

Figure 1: Results of **SAC+HER** on 6 selected environments. Note that these plots mainly show the environments on which VDIFF-R improves performance. Complete results can be found in Figure 4. Shaded region represents 95% confidence interval of mean success rate across 6 seeds.
.

- **VDS:** VDS (Zhang et al., 2020) is the state-of-the-art curriculum method for goal-conditioned RL. It proposes to sample goals with high epistemic uncertainty which is approximated as the disagreement between the outputs of an ensemble of learned value functions. Our implementation follows the official codebase of VDS. However, we implement VDS with SAC instead of DDPG (Lillicrap et al., 2015) to enable fairer comparison. This is not an issue as VDS is independent of the base RL algorithm and treats it as a black box.

### 4.2 IMPROVEMENTS USING VDIFF AND RESAMPLING

Results of all methods using HER+SAC are presented in Figure 1. In the following discussion, we drop '+HER' from the names of the explicit curriculum algorithms for brevity. For example, VDIFF actually refers to VDIFF+HER. From the results, we first observe that VDIFF-R shows improved sample efficiency or better performance compared to baseline methods in 7 out of the 17 environments. In the other 10 environments, VDIFF-R matches the performance of baseline methods. Notably, the performance gains are significant in 3 variants of the challenging *Hand Manipulate Block* environments. By comparing VDIFF and VDIFF-R, we can conclude that continuous goal sampling is responsible for the observed gains.

We hypothesize that there could be 2 major reasons behind the ineffectiveness of VDS and VDIFF when used alongside HER. First, HER already has a powerful implicit curriculum because as an agent's policy improves, the goals it uses for replay naturally shift from simpler to more difficult ones. As it generates training data for the agent, it also has a stronger and more direct impact on an agent's learning. The potential gain of using an explicit curriculum method alongside HER could thus be small. Second, HER augments the buffer through fictitious data which is often biased. Since both VDS and VDIFF use value functions trained on this buffer, their generated curricula might be noisy and biased. At the same time, it is worth noting that while HER is constrained to work only in sparse reward settings and with off-policy methods, both VDS and VDIFF have no such restrictions. Hence, it is crucial to understand their performance without HER, which is discussed in Section 4.4.

### 4.3 CAN RESAMPLING HELP OTHER ALGORITHMS?

Figure 2 presents results comparing continuous resampling variants of the baseline methods to their vanilla variants. In the following discussion, we drop '+HER' from the names of the explicit curriculum algorithms for brevity. For example, VDS-R actually refers to VDS-R+HER. We first observe that both VDS-R and HER-R outperform their corresponding vanilla variants, VDS and HER on

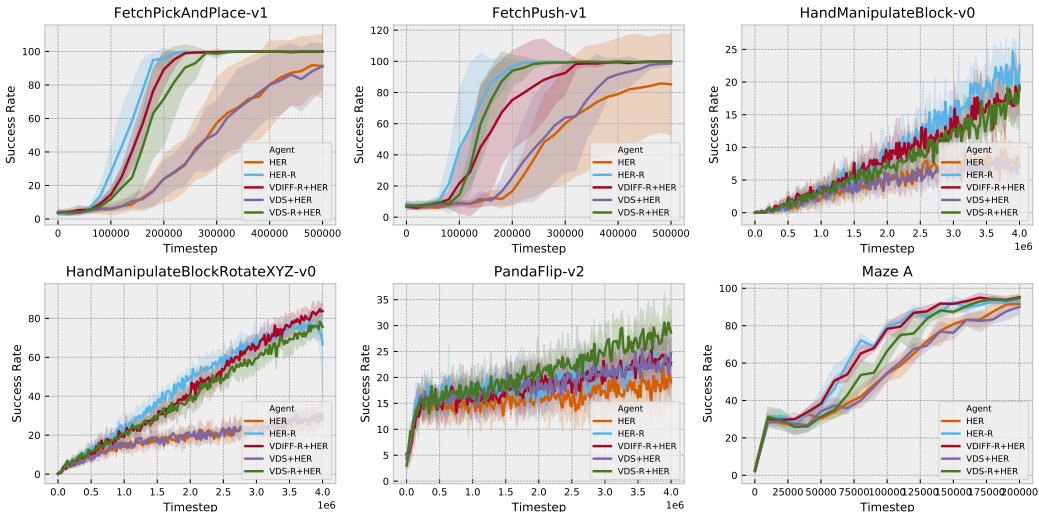

Figure 2: Results for continuous goal sampling variants of HER and VDS on 6 selected environments. Note that these plots mainly show the environments on which continuous goal sampling improves performance over baseline methods. Complete results can be found in Figure 5. Shaded region represents 95% confidence interval of mean success rate across 6 seeds.

6 environments. In the other 11 environments, they match the performances of VDS and HER. This provides further evidence that resampling goals is a general strategy which can accelerate performance in goal-conditioned RL. Interestingly, we also observe that HER-R is able to marginally outperform VDS-R and VDIFF-R. Since HER-R always resamples goals while VDS-R and VDIFF-R only selectively resample, this leads us to the conclusions that i) random resampling at a constant frequency is an effective strategy with HER and ii) explicit curriculum methods might have lesser room for improvement in settings with HER (further validated through results in Section 4.4).

## 4.4 IS VDIFF EFFECTIVE WITHOUT HER?

In Figure 3, we compare all baseline methods on SAC without HER. We first observe that both VDS and VDIFF are easily able to outperform Random in 6 out of the 12 tested environments. Without the strong implicit curriculum of HER, the role of explicit curriculum methods increases greatly. It is also worth noting that VDIFF is able to outperform VDS on multiple environments. Next, and as expected, continuous goal sampling helps further improve the performance of all methods. Finally, it is also interesting to note that not all environments are amenable to learning with HER. For example, we observe that though methods using HER generally perform significantly better in most environments, their asymptomatic success rate is lesser than equivalent NO-HER variants for 3 environments (*Hand Manipulate Pen,Hand Manipulate Pen-Rotate and Fetch Slide*). This is possibly because of the biased data that HER adds into the replay buffer. In environments where the performance is affected by such bias, using explicit curriculum learning methods with no HER is the best choice.

## 4.5 HOW DOES RESAMPLING FREQUENCY AFFECT PERFORMANCE?

To better understand what the optimum resampling frequency $(R)$ is, we run an ablation study over different values for $R$. These results are depicted in Figure 7. Note that $R = \textit{Episode Length}$ represents the standard practice of sampling a new goal when the environment is reset. From these results, we first observe that performance is roughly the same for moderate values of $R$, i.e., $R = 10, 20$. However, performance becomes increasingly unstable as $R$ is decreased further and is the worst for the extreme value of $R = 1$. This is expected for two reasons. First and intuitively, sampling a new goal at every timestep breaks the sequential decision-making nature of the problem. Second and more technically, if a new goal is sampled at some timestep $t$, then the $(s_t, a_t, s'_t)$ transition tuple is unsuitable for temporal difference (TD) updates as the reward function has changed. For higher

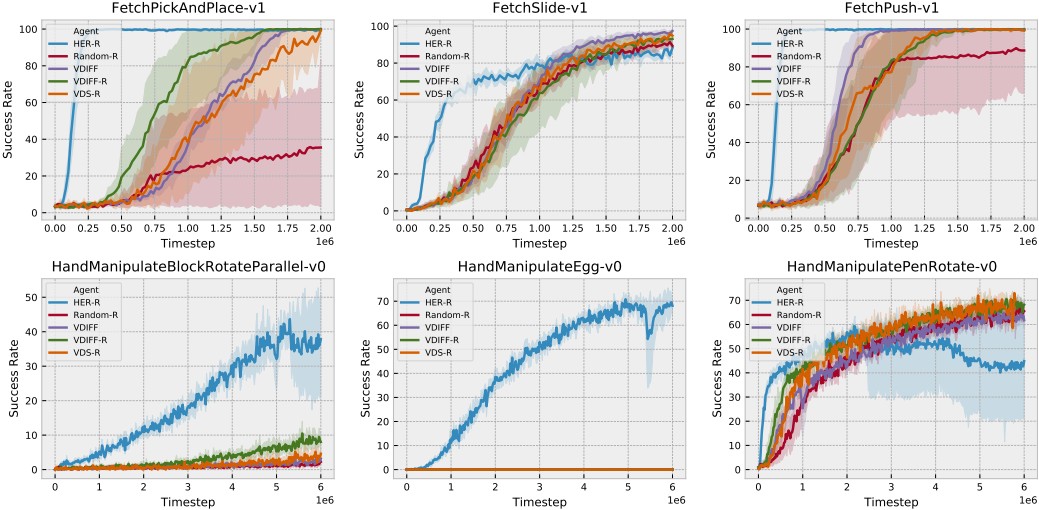

Figure 3: Results **without HER** on all environments. Shaded region represents 95% confidence of mean success rate across 6 seeds. We observe that the explicit curriculum methods VDS and VDIFF easily outperform Random. However, HER is able to outperform other methods on most environments. Interestingly, there are 3 environments in which HER converges to a lower success rate compared to VDIFF-R. Full results can be found in Figure 6.

values of $R$, this does not matter as such transitions are relatively rare (even rarer when there is data augmentation from HER which is free of this issue). However, it will still be interesting to study if removing these rare erroneous transitions from the buffer can further improve learning. We find that setting $R = 0.2 \times MaxEpisodeLength$ generally works well for most environments.

### 4.6 HOW DOES THE TARGET VALUE FUNCTION OF VDIFF IMPACT PERFORMANCE?

VDIFF uses target value functions to compute learning progress (LP) (Equation 2). To understand the impact of using the target function, we run an ablation study in which LP is instead computed using the standard (non polyak-averaged) value functions, which were described in Equation 1. To minimize additional sources of noise and to bring full focus on the explicit curriculum method, we run these experiments without HER. In addition to the success rate, we also plot the estimated LP, which is computed using the appropriate value functions, as described above. The obtained results are depicted in Figure 8. From the results, we observe that standard value functions perform worse compared to target value functions in multiple environments. As discussed in Section 3.2.1, we believe that that this is a result of the noisiness of the value function. This noise causes significant variation in the LP estimates, which can be directly seen from the LP plots.

## 5 RELATED WORK

### 5.1 ORIGIN OF CURRICULUM LEARNING

The origins of curriculum learning are rooted in human and animal cognition. Early experiments showed that animals could be trained more effectively by administering a curriculum of programmed learning (also referred to as shaping in cognitive science) (Ferster & Skinner, 1957). Inspired by this, Bengio et al. (2009) conceptualized curriculum learning in the context of machine learning by training a supervised learning algorithm on examples of gradually increasing difficulty. This was soon followed by works which developed an *active* (adaptive) curriculum for supervised learning algorithms (Kumar et al., 2010; Lee & Grauman, 2011; Graves et al., 2017). Due to its parallels with learning in biological agents, RL is one of the most promising domains to apply curriculum learning. With the advent of deep RL, there has been increased interest in developing more general-purpose RL agents that can solve multiple tasks (Narvekar et al., 2020; Soviany et al., 2022).

## 5.2 CURRICULUM LEARNING IN GOAL-CONDITIONED RL

Curriculum learning for goal-conditioned RL (GCRL) is concerned with the problem of presenting training goals to an agent in an order which is conducive to learning (Schaul et al., 2015; Liu et al., 2022). Florensa et al. (2018) trained a generative adversarial network (GAN) to generate goals that have an intermediate probability of success. Zhang et al. (2020) proposed to sample goals which have high epistemic uncertainty (Section 4.1). Hindsight Experience Replay (HER) is a popular relabeling technique that uses future states reached in a trajectory as virtual goals to augment the replay buffer (Andrychowicz et al., 2017). By doing so, it builds an implicit curriculum for the agent as goals used for replay naturally shift from simpler to more difficult ones. Curriculum-guided HER (CHER) is an extension of HER which uses curriculum learning to select the virtual goals used by HER for relabeling (Fang et al., 2019). Finally, self-play based methods have a multi-agent setup (generally 2 agents) in which one agent proposes increasingly challenging yet achievable goals for the other agent (OpenAI et al., 2021; Sukhbaatar et al., 2017; Du et al., 2022; Campero et al., 2020). Although powerful, self-play based methods are primarily targeted towards open-ended environments, and require a complex setup and a large number of training samples.

## 5.3 LEARNING PROGRESS BASED CURRICULUM

The fundamental idea behind learning progress (LP) based curriculum methods is to sample with high frequency the goals/tasks on which an agent is making the most progress. LP can be considered a form of intrinsically motivated active learning and has been used successfully in diverse applications ranging from developmental robotics (Blank et al., 2005) to classroom teaching (Clement et al., 2013). R-IAC, one of the first active learning algorithms to use LP in sensorimotor learning, used absolute LP (ALP) to split a parameter space into sub-regions (Baranes & Oudeyer, 2009). Building on this, Matiisen et al. (2019) presented a teacher-student framework for discrete task spaces in which a teacher (curriculum generator) samples tasks with high LP for the student (RL agent). Portelas et al. (2020) extended this to continuous task spaces by fitting a Gaussian Mixture Model (GMM) on (task, ALP) tuples and then sampling a task from a Gaussian chosen proportionally to its mean ALP value. Both these works assume dense reward settings and compute LP using either the nearest neighbor (Portelas et al., 2020) or the change in episode return for a given task (only applicable in discrete task spaces). As a result, they are not directly applicable to GCRL, which not only has binary episode returns, but also has random initial states that are possibly high dimensional. This work reformulates ALP for GCRL using value functions, as described in Section 3.2.1. Similar to our method, SPaCE also uses value functions for LP computation (Eimer et al., 2021). However, SPaCE is targeted towards contextual RL and is only applicable in discrete task settings. Additionally, it does not make use of target value functions for smoothing.

## 6 CONCLUSION AND FUTURE WORK

In this work, we presented 2 contributions to advance goal-conditioned RL. First, we proposed continuous goal sampling, a novel extension of standard GCRL in which goals for an agent are sampled and set multiple times during an episode. Second, we introduced VDIFF, an automatic curriculum learning method for GCRL, which uses learning progress computed through value functions to develop a self-paced curriculum for an agent. Through experiments on a suite of 17 sparse-reward tasks, we demonstrated the effectiveness of continuous goal sampling as a general strategy to accelerate learning in GCRL. Finally, we showed that although explicit curriculum methods such as VDIFF can generate meaningful curricula, their role is minimized in settings where Hindsight Experience Replay (HER) (Andrychowicz et al., 2017) can be applied.

Future work will explore different variants and applications of continuous goal sampling. This could include exploring techniques such as dynamically changing episode length and applications such as reset-free RL (Zhu et al., 2020). Another interesting future direction will be to better understand the interaction between explicit curriculum methods and HER, with the ultimate intent of developing explicit curriculum methods that can work harmoniously with HER. Finally, it will also be interesting to evaluate explicit curriculum methods such as VDIFF on on-policy algorithms and dense reward settings, and study techniques that can extend VDIFF to multi-task RL (Romac et al., 2021).

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

## A APPENDIX

### A.1 IMPLEMENTATIONAL DETAILS

This section presents implementational details and minor modifications which enable continuous goal resampling and VDIFF to work effectively in practice.

**Sampling Goals:** Even though we can easily obtain $LP(s_t, g)$ for any $(s_t, g)$, computing $p^\pi(g|s_t)$ over the entire continuous goal space is intractable. To enable sampling from $p^\pi(g|s_t)$, we discretize the problem by first sampling a set of goals $\mathbb{G} = \{g^{(n)}\}_{n=1}^N$ using $U(g)$, the uniform distribution over the goal space $G$. Then, the probability of sampling goal $g \in \mathbb{G}$ is

$$p^\pi(g|s_t) = \frac{1}{Z} f(s_t, g) \tag{6}$$

where $Z = \sum_{i=1}^N f(s_t, g^{(i)})$ is a normalizing constant, and $f(s_t, g)$ is the function encoding the curriculum objective (in the case of VDIFF, $f(s_t, g) = LP(s_t, g)$).

**Catastrophic Forgetting:** Multi-task RL is prone to catastrophic forgetting, a phenomenon where agents forget how to achieve learned goals if the learned goals are not sampled enough or if the agent's policy changes significantly in the process of learning new goals . This issue can be exacerbated by curriculum learning methods because they alter $p(g|s_t)$ which could result in some goals having a very low probability of being sampled. After initial early training for $M$ episodes with LP objective, we samples goals using the absolute learning progress (ALP) -

$$p^\pi(g|s_t) \propto |LP(s_t, g)| \tag{7}$$

We do not use ALP in the early training phase because value functions are initialized randomly to output values close to $0$. Given that our reward structure has negative rewards, outputs of the value function fall during early training. If we use ALP as an objective from the start, the drop in output values might be falsely detected as catastrophic forgetting, when in fact it is the initial convergence of the value function. Finally, in actual practice, we find that agents in multi-goal RL are less prone to catastrophic forgetting in comparison to multi-task RL (Portelas et al., 2020; Matiisen et al., 2019). This is expected as multi-task RL considers a distribution of environments, whereas multi-goal RL is more structured and involves a distribution over sparse reward functions in a single environment.

**Efficient Updating of V':** Since updating $V^{T'}$ at each training step can be expensive, we sync it with $V^T$ every $F$ episodes, where $F$ is referred to as the *fit rate*.

### A.2 ALGORITHMS
### A.2.1 SELECTIVE GOAL RESAMPLING

---
**Algorithm 3** Selective Goal Resampling Algorithm

---
  **Input:** Max Episode Length $T_{max}$, Resample Frequency $R$, policy $\pi$
  **Initialize:** Initial state $s_0 \sim \rho(s_0)$, Initial goal $g_{current} \sim p^\pi(g|s_0)$ [Using equation 6]
  **for** $t = 0, 1, ...T_{max}$ **do**
    $a_t \sim \pi(s_t, g_{current})$
    $s_{t+1}, r_{t+1}, done = environment\_step(a_t)$
    **if** $done$ **then**
      $break$
    **end if**
    **if** $t$ **mod** $R == 0$ **then**
      Sample set of $N$ random goals $\mathbb{G}$ using $U(g)$
      $\mathbb{E}_\mathbb{G}[f(s_t, g)] = \sum_{i=1}^N [p(s_t, g^{(i)}) \times f(s_t, g^{(i)})]$   []Equation 6
      **if** $E_\mathbb{G}[f(s_t, g)] > f(s_t, g_{current})$ **then**
        $g' \sim p^\pi(g'|s_{t+1})$
        $g_{current} = g'$
      **end if**
    **end if**
  **end for**

---

## A.3 COMPLETE EXPERIMENTAL RESULTS

### A.3.1 IMPROVEMENTS USING VDIFF AND CONTINUOUS GOAL SAMPLING

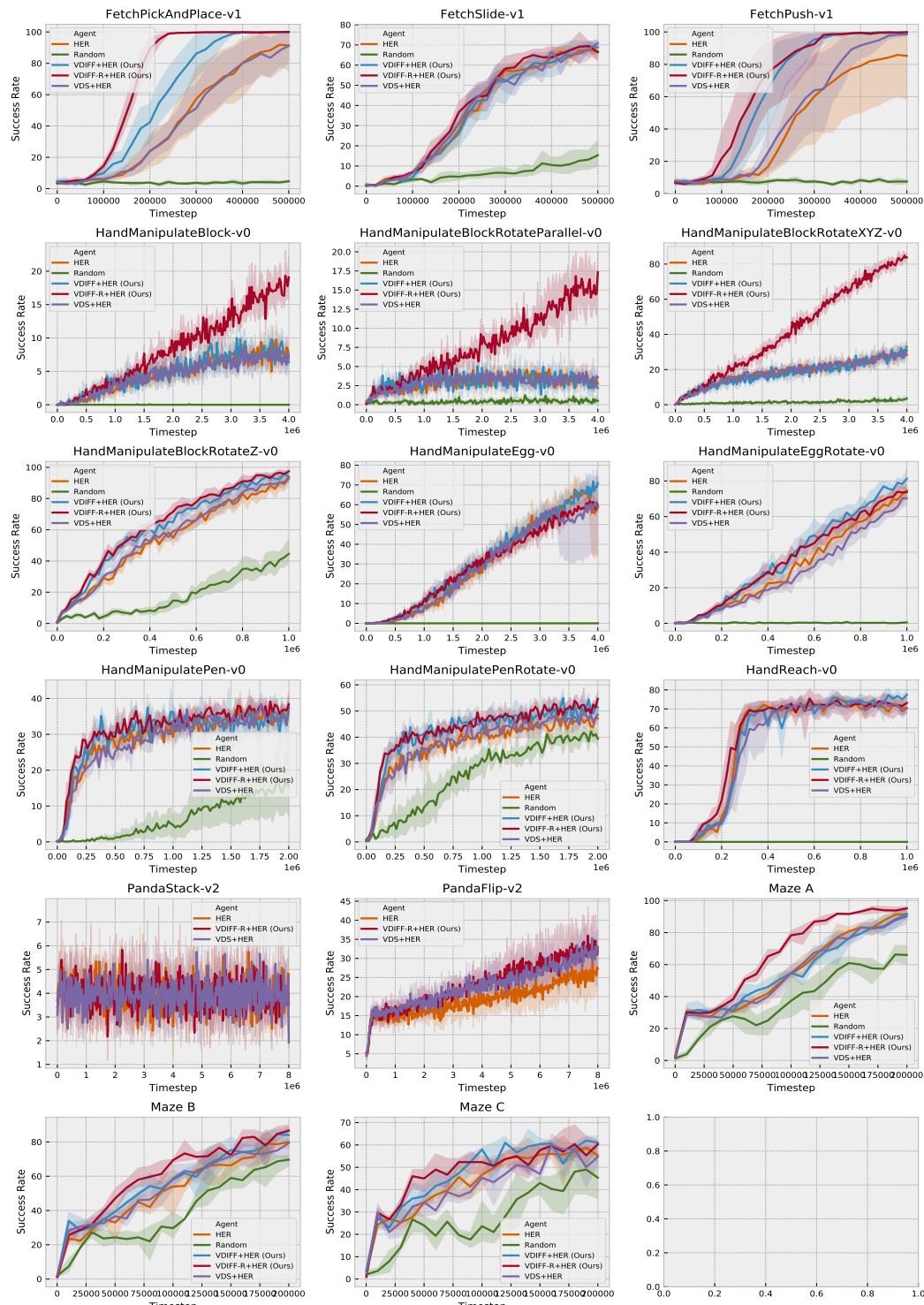

Figure 4: Results for SAC+HER on all environments. Shaded region represents 95% confidence interval across 6 seeds. We notice that VDIFF-R improves performance in 7 environments and matches the performances of baseline methods in the other 10 environments.

.

A.3.2   EFFECT OF CONTINUOUS GOAL SAMPLING ON BASELINE ALGORITHMS

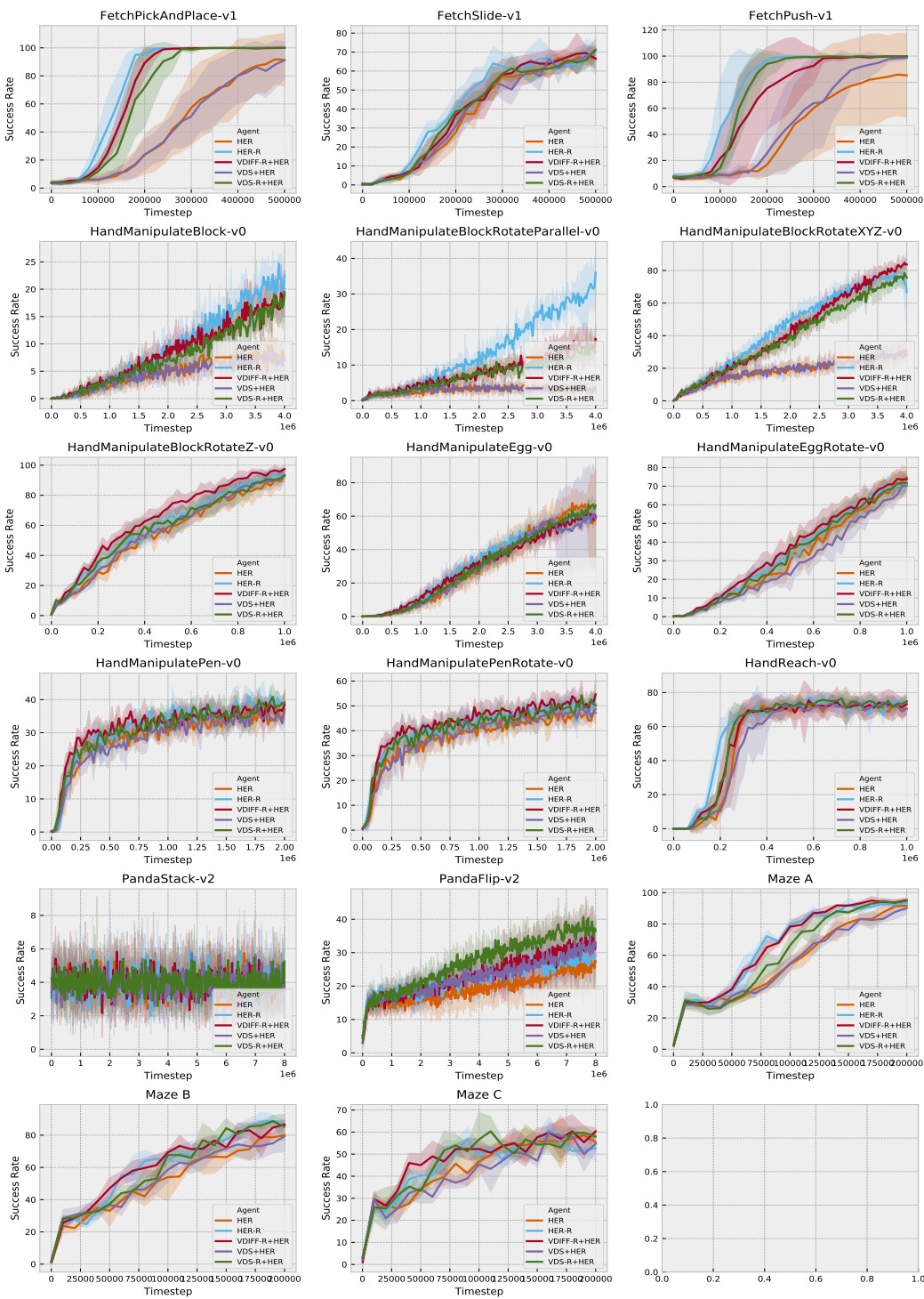

Figure 5: Results showing continuous goal sampling for HER and VDS on all environments. Shaded region represents 95% confidence interval of mean success rate across 6 seeds. We notice that the continuous sampling variants HER-R and VDS-R outperform their corresponding vanilla variants in multiple environments.
.

### A.3.3   RESULTS IN THE NO-HER DOMAIN

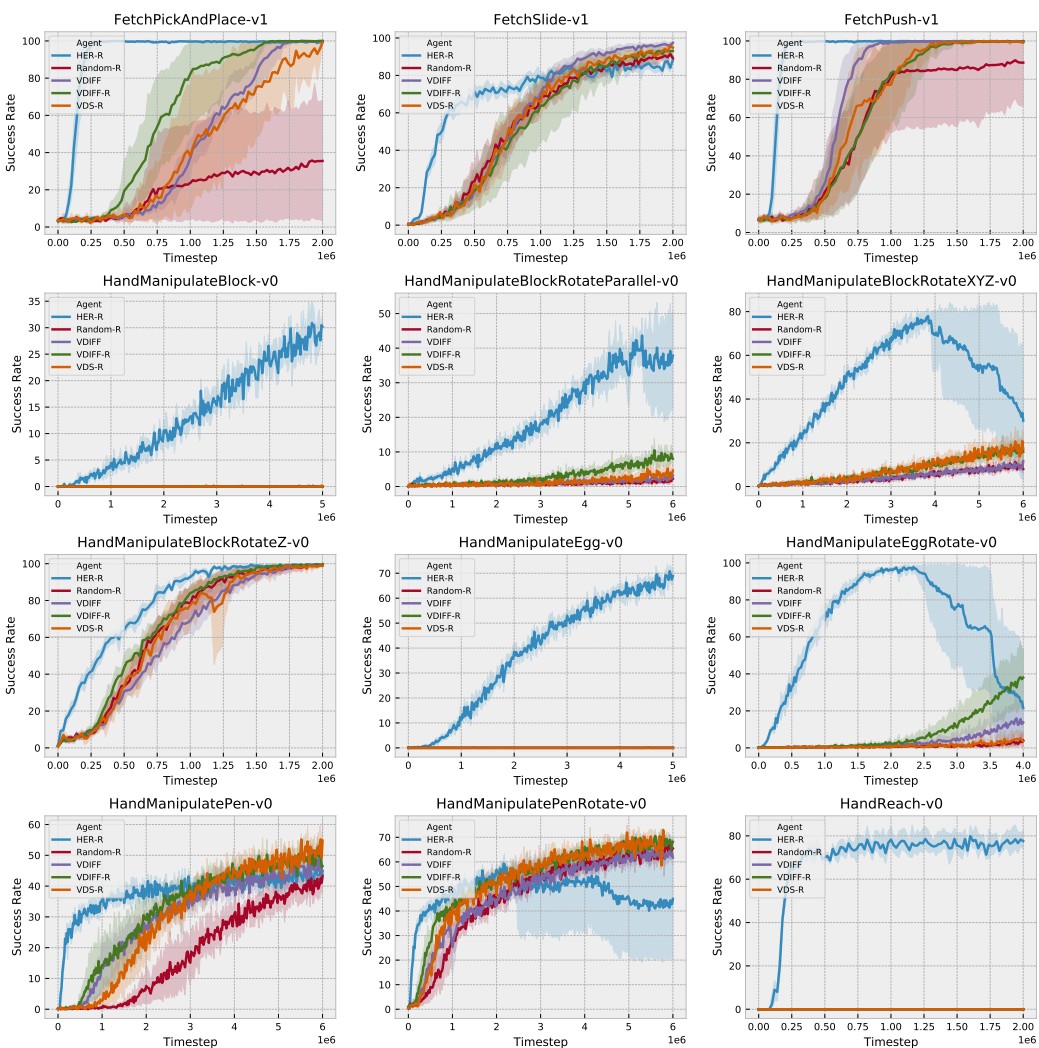

Figure 6: Results **without HER** on all environments. Shaded region represents 95% confidence of mean success rate across 6 seeds. We observe that the explicit curriculum methods VDS and VDIFF easily outperform Random. However, HER is able to outperform other methods on most environments. Interestingly, there are 3 environments in which HER converges to a lower success rate compared to VDIFF (*Hand Manipulate Pen,Hand Manipulate Pen-Rotate and Fetch Slide*). Note that in some environments, HER crashes after converging. Although this does not matter in practice as we save/pick the model at the best success rate, it is an example of an adverse effect caused by the bias of HER.

### A.3.4 ABLATION ON RESAMPLING FREQUENCY

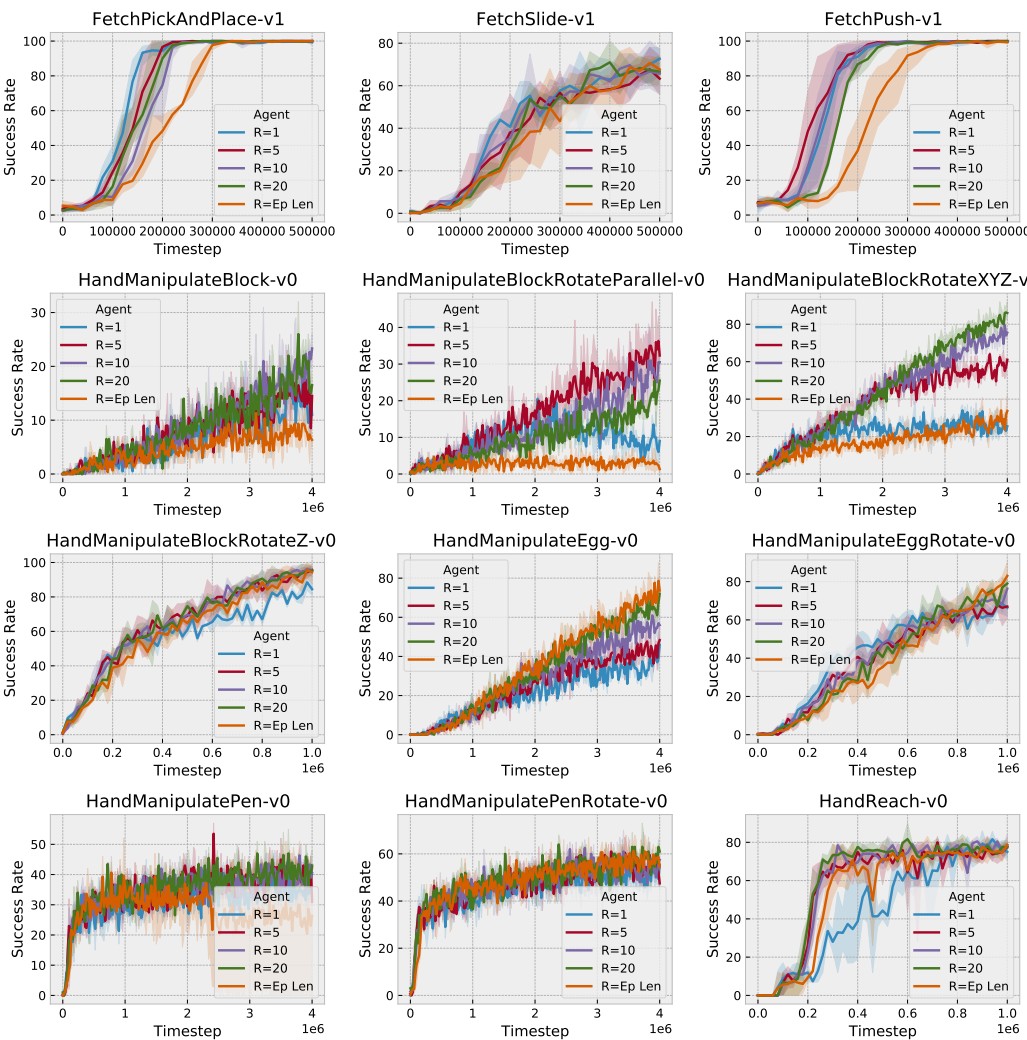

Figure 7: Ablation analysis on the choice of resample frequency $R$ for all environments. All methods use VDIFF+HER with identical hyperparameters. $R = Ep\ Len$ represents the standard practice of sampling a new goal when the environment is reset. Shaded region represents 95% confidence interval of mean success rate across 3 seeds. We observe that performance becomes increasingly unstable as $R$ is decreased.

### A.3.5 ABLATION ON THE TARGET VALUE FUNCTION OF VDIFF

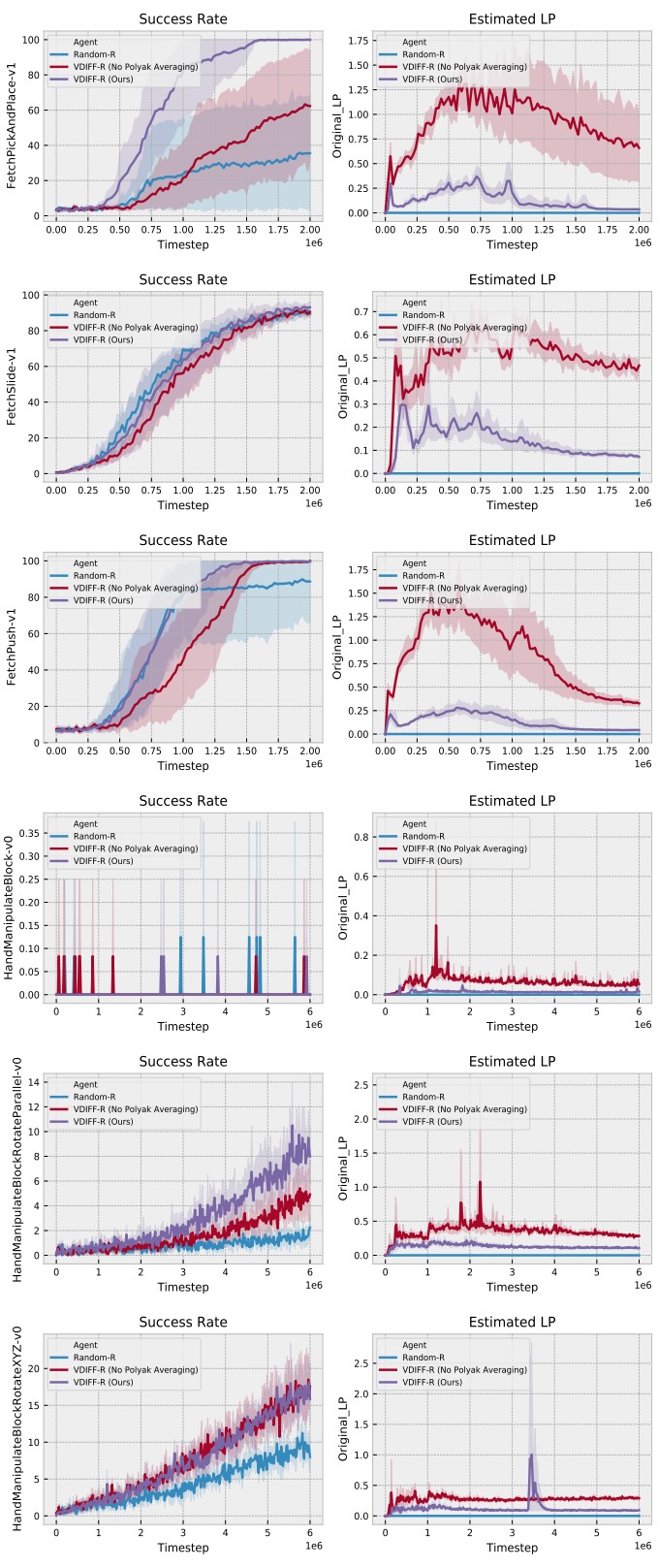

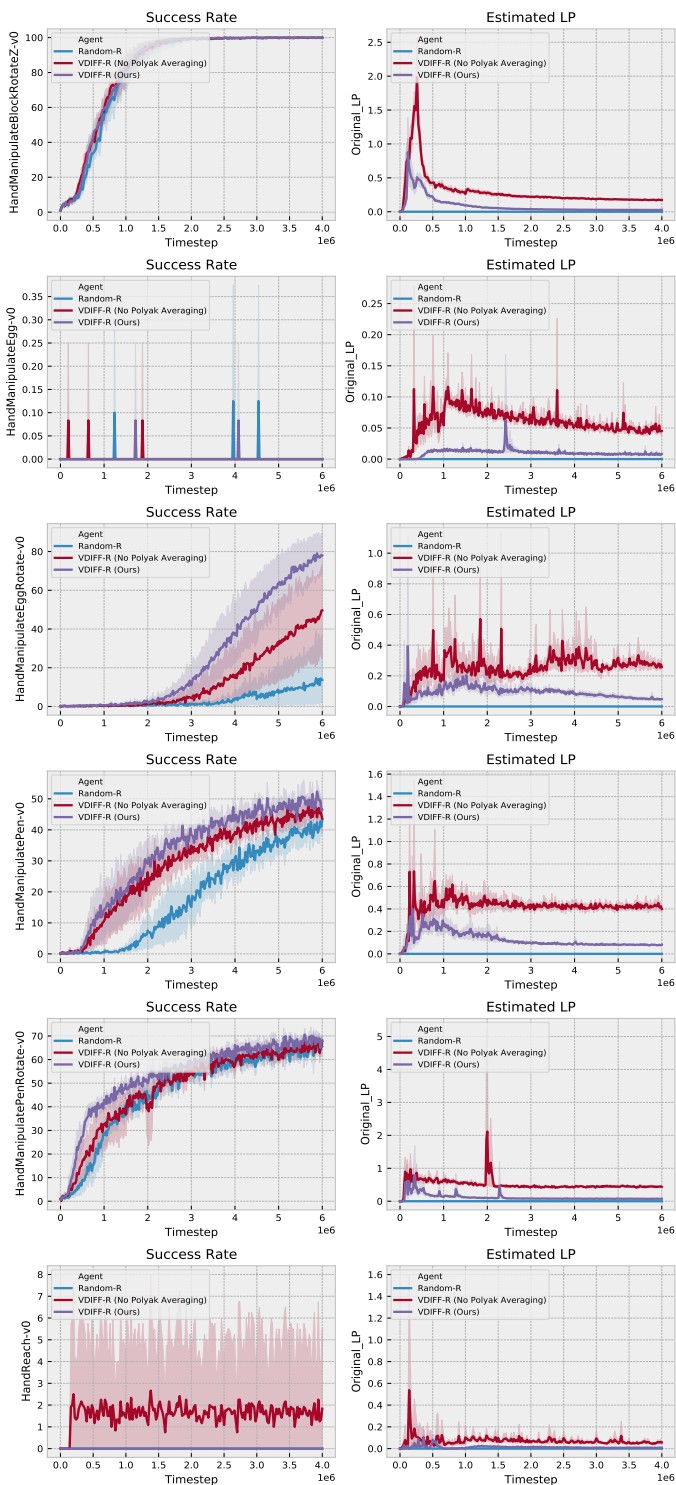

Figure 8: Ablation analysis on the target value functions of VDIFF. Shaded region represents 95% confidence interval of mean success rate or mean estimated LP across 6 seeds. We observe that standard value functions lead to noisier estimated LP and perform worse compared to target value functions in multiple environments.

