# OpenReview forum: "Continuous Goal Sampling: A Simple Technique to Accelerate Automatic Curriculum Learning"
_ICLR.cc/2023/Conference — Submitted to ICLR 2023_

### Official Review · Reviewer_UKQZ · 2022-10-24

**Confidence:** 4
**Correctness:** 4
**Technical Novelty And Significance:** 3
**Empirical Novelty And Significance:** Not applicable
**Recommendation:** 6

**Clarity, Quality, Novelty And Reproducibility:**

I find the writing clear, and the results to be of high quality. As mentioned above, I find the continuous sampling part to be novel (yet simple), whereas the learning progress part of the contribution to be only marginally novel compared to SPaCE. I find the results reproducible.

**Strength And Weaknesses:**

## Strengths
Continuous goal sampling is a simple, but a reasonable idea. It fits well within the Zone of Proximal Development (ZPD) framework[1] which states that tasks presented to the agent should be neither too easy nor too difficult. I see the standard "termination at the reward" as satisfying the "not too easy" part of the ZPD motivation, and authors "change the goal every R steps" naturally rejecting "too hard" tasks (if there is no reward in R steps, the goal gets rejected).

The experiments are performed in many (current, external to the paper) domains and the baselines of variants of VDS and HER are chosen appropriately. The paper tells a coherent story, supported by many experimental results.

## Weaknesses
Learning progress seems to only differ from SPaCE through the use of Polyak averaging. Authors claim that "SPaCE is targeted towards contextual RL and is only applicable in discrete task settings.", however I think SPaCE could be a reasonable additional baseline to try. Alternatively, the addition of Polyak averaging could be ablated, to understand the impact of the novel part proposed by this paper.

## Smaller suggestions

I would like to see HER performance on the Figure 3. Authors suggest that in many tasks, a further progress is difficult compared to a baseline of HER; it would be good to confirm that (where) the gains of HER are reproduced by VDIFF. It would make it clear whether the conclusion is:

1. Sometimes HER is better, sometimes their method, best to use both, or
2. VDIFF (with no HER) is always better, but sometimes HER is enough.

## Longer-term, future work suggestions

I find changing goals without resetting the environment only partially symmetric to "finishing episode at the reward"[2], as ending the episode on reward resets the environment, whereas for changing the goal the env is not reset. There are two alternatives which would treat both constraints in the same way:

1. Change "goal sampling" to reset the environment when there is no reward in R steps, or
2.  Don't reset the environment at the reward. Rather, continue running the environment, but cut the bootstrapping beyond the first reward, to make the agent only optimize "time until the first reward" and not keeping the reward satisfied later.

It's unclear how the proposed method deals with the situation where some tasks are inherently harder (=> require more time to solve optimally) than others, and thus require different Rs.

[1] Vygotsky, [Interaction Between Learning and Development](https://innovation.umn.edu/igdi/wp-content/uploads/sites/37/2018/08/Interaction_Between_Learning_and_Development.pdf)

[2] I understand authors made no claims to the symmetry here, but for the future work, I find it interesting to analyse the situation in more detail, which would make the results even stronger in my mind.

**Summary Of The Paper:**

The paper proposed two methods to improve auto curriculum learning strategies in the RL settings where multiple goals are available with no change to the environment dynamics:

1. Continuous goal sampling, where the goal of the agent changes every R < episode length steps
2. A curriculum based on "learning progress", which chooses to sample tasks (described here by a tuple (initial_state, goal) proportionally to the improvement in the value function within dT episodes, and to smooth these VFs using Polyak averaging.

The method is evaluated on a number of manipulation and maze navigation tasks, with the performance compared with VDS and HER. In the basic version, authors propose to use their method (called VDIFF) together with HER, but they also provide ablations on the effect of using HER with the conclusions that in some of the tasks VDIFF improves over HER and VDS, and in others VDIFF makes the same gains (ie. performance of HER is comparable to VDIFF which is better than the lack of both).

**Summary Of The Review:**

The authors propose two improvements to the curriculum learning for goal-conditoned RL. One of them is simple but novel, the other one is a marginal improvement over previous work (SPaCE). The experimental results are solid and convincing.

I think the goal sampling idea would be an interesting addition to the conference, so I lean towards acceptance. The strength of my sentiment is weak due to the second, less novel part and the fact that the idea is something many would have tried even if they didn't read the paper (yet I haven't managed to locate it among previous literature).

---

> ### Author Response · Authors · 2022-11-14
> **Response to Official Review of Paper3535 by Reviewer UKQZ**
>
> We thank you for the time and effort you put into this review, and we are glad that you think this paper would be an interesting addition to the conference. We provide clarifications for your questions below -
>
> >**Learning progress seems to only differ from SPaCE through the use of Polyak averaging. SPaCE could be a reasonable additional baseline to try. Alternatively, the addition of Polyak averaging could be ablated, to understand the impact of the novel part proposed.**
>
> We present additional results to empirically validate the usefulness of target value functions. As suggested by the reviewer, we ablate the impact of polyak averaging. While this is not a reimplementation of SPaCE [1], it is very close to the definition of LP which is used by SPaCE. More information can be found in the global [comment][here].
>
> >**I would like to see HER performance on Figure 3. Authors suggest that in many tasks, further progress is difficult compared to a baseline of HER.  It would be good to confirm that (where) the gains of HER are reproduced by VDIFF.**
>
> We have added HER to Figure 3 as requested. For clarity, in most environments, HER-R (or any method using HER and resampling) has the best performance and should be the go-to method. In these environments, VDIFF (without HER) is unable to match HER in performance or sample efficiency. We apologize if this was not clear before and have added sentences in the paper to make this more explicit. At the same time, there are a few environments in which VDIFF (without HER) outperforms all HER-based methods, possibly due to the bias introduced by HER. If a practitioner only cares about the final success rate, then VDIFF-R (No HER) could be useful in such environments.
>
> In general, we believe that VDIFF-R (with HER) or HER-R should be the first methods tried on any given goal-conditioned environment. There are also a few additional settings which fall within the scope of VDIFF but outside the scope of HER. These have been discussed in another [comment][her].
>
> >**Future Work/ Longer-term suggestions: I find changing goals without resetting the environment only partially symmetric to "finishing episode at the reward"[2], as ending the episode on reward resets the environment, whereas for changing the goal the env is not reset. There are two alternatives which would treat both constraints in the same way. It's unclear how the proposed method deals with the situation where some tasks are inherently harder (=> require more time to solve optimally) than others, and thus require different Rs.**
>
> Thank you for this insightful analysis! Yes, both suggested alternatives deserve careful analysis in future work. We give general thoughts on the suggested variants -
>
> - Based on our understanding of your first suggestion, resetting the environment every R steps is equivalent to having a maximum episode length of R. This is certainly plausible but might affect performance (for better or for worse) in two ways. First, HER uses future goals of a trajectory for goal substitution. Having shorter episodes will affect the distribution of future goals sampled for HER. Second, if the environment always resets to a fixed initial state distribution, then having shorter episode lengths might restrict exploration as frequent resetting would imply that agents spend more time close to initial states.
>
> - Based on our understanding of your second suggestion, the environment should run for a fixed episode length and multiple goals can be achieved by the agent in the same episode (the returns of the trajectory will be appropriately modified). Again, it would be interesting to see how this variant interacts with HER and how it affects exploration. We believe that 2) might lead to better performance than 1), but 1) is simpler to integrate into existing codebases.
>
> Finally, having an adaptive R will be another interesting direction to explore. While we currently selectively resample goals which is partially equivalent to having an adaptive R, this does not seem to have a major impact on performance. However, we believe that there is scope for improvement here, both by considering better objectives for selective resampling and by exploring techniques to adaptively vary R. Once again, we thank you for engaging into an interesting discussion about this!
>
> We hope we have been able to address your concerns in the above clarifications. Kindly let us know if you have additional questions and we would be more than happy to discuss further.
>
> [1] : Eimer, Theresa, et al. "Self-paced context evaluation for contextual reinforcement learning." International Conference on Machine Learning. PMLR, 2021.
>
> [2] : Plappert, Matthias, et al. "Multi-goal reinforcement learning: Challenging robotics environments and request for research." arXiv preprint arXiv:1802.09464 (2018).
>
> [here]:https://openreview.net/forum?id=Vk9RH9aL1Yv&noteId=Oo-2o-YHTe
>
> [her]:https://openreview.net/forum?id=Vk9RH9aL1Yv&noteId=GlG_hZJaiS

---

> > ### Comment · Reviewer_UKQZ · 2022-12-05
> > **short update**
> >
> > After discussion with other reviewers and looking through the changed manuscript I grew the following opinions:
> > + now that selective resampling (which I missed in the first read) is part of the main text, it seems reasonable to join the two methods together in a single paper (as opposed to having two separate papers for the method)
> > - given the idea of resampling is simple, the community won't lose that much if it were not published immediately, as a practitioner may try a variation of it whether the paper is published on not. Thus, the experimental standards for validation of the paper's claims should be set relatively high, and the current somewhat mixed experimental results are not convincing enough
> > - the paper would benefit a lot from a quantitative experiment which could help the reader to build an intuition on how the method works (and what are its limitations) in a reduced scale.
> >
> > As such, I believe the paper could have a significantly bigger impact had these issues be addressed before publication; my current recommendation is to not accept.

---

### Official Review · Reviewer_aTaV · 2022-10-24

**Confidence:** 3
**Correctness:** 4
**Technical Novelty And Significance:** 2
**Empirical Novelty And Significance:** 2
**Recommendation:** 3

**Clarity, Quality, Novelty And Reproducibility:**

I think the paper is written in ways that could be misleading. Writing should be clarified and improved. I have no concern about reproducibility. Resampling goals has been used previously by practitioners designing tasks to elicit particular behaviours but to my knowledge it has not been explicitly incorporated into an algorithm previously. Similarly learning progress has been considered for curriculum generation but not in this exact formulation.

**Strength And Weaknesses:**

Strength:
* goal resampling seems like a generally applicable and useful idea.
* VDIFF-R is empirically promising in the setting without HER

Weaknesses:
* There are two algorithmic contributions that are orthogonal. The first figure is misleading and I would prefer it if the results of figures 1 & 2 are combined in one figure. Since the resampling can be applied to other methods those should be present in the first comparison
* Related to the first weakness, I think the abstract is misleading and contains not enough nuance. "continuous goal sampling and VDIFF work synergistically and result in performance gains over current state-of-the-art methods" does not seem quite right when VDIFF-R does not outperform HER-R or VDS-R.
* I am not particularly convinced that settings without HER are very interesting. What is a setting where HER cannot be used? If that is where VDIFF shines then there should be an evaluation in such a setting directly rather than evaluating in a setting where HER can be used.
* I think it would be good to include the algorithms in the main text if possible. Similarly, I think the selective resampling mentioned in the appendix ought to be mentioned in the main text as it adds complexity.

**Summary Of The Paper:**

This paper focuses on goal conditioned RL and introduces two algorithmic contributions:
* goal resampling within an episode (called continuous goal resampling)
* a automatic curriculum learning method called VDIFF that samples appropriate goals based on learning progress (defined as the change in predicted (target) value for the same state and goal after some number of updates)
The paper demonstrates empirically that goal resampling is beneficial for VDIFF and other goal-conditioned RL methods. VDIFF is similar to other methods when hindsight experience replay (HER) is used but performs better without HER.

**Summary Of The Review:**

The paper proposes two orthogonal contributions. To my mind it is written in a slightly misleading way and does not do enough to justify why the setting in which VDIFF-R performs well (no HER) is interesting and relevant. If there is an interesting setting where HER cannot be used then the evaluation should at least partially be in that setting.

---

> ### Author Response · Authors · 2022-11-14
> **Response to Official Review of Paper3535 by Reviewer aTaV**
>
> We thank you for the time and effort you put into this review, it has greatly helped us understand how we can improve our paper. We provide clarifications and comments for your questions below -
>
> >**There are two algorithmic contributions that are orthogonal. The first figure is misleading and I would prefer it if the results of figures 1 & 2 are combined in one figure.**
>
> The main purpose of this work was to introduce two simple ideas that an RL practitioner can easily integrate into their goal-based RL framework. While the two ideas are not directly correlated, we believe that both of them could help improve performance in general goal-conditioned settings. Figures 1 and 2 were separated both for a smoother flow to the story and for clarity. The discussion of Figure 1 (Section 4.2) acknowledges that the gains of VDIFF in settings with HER are minimal. Combining the two figures will lead to having 7 plots on the same graph, which would affect readability.
>
> >**Related to the first weakness, I think the abstract is misleading and contains not enough nuance. "continuous goal sampling and VDIFF work synergistically and result in performance gains over current state-of-the-art methods" does not seem quite right when VDIFF-R does not outperform HER-R or VDS-R.**
>
> Thank you for pointing this out. We agree that the highlighted statement could be misleading and have modified the sentence to more clearly reflect the actual picture. In general, we have tried to increase the clarity of writing in different sections of the paper. For example, the last sentence of the abstract now reads -
>
> *‘Through results on 17 multi-goal robotic environments and navigation tasks, we show that continuous goal sampling, combined with VDIFF or existing curriculum learning methods, results in performance gains over state-of-the-art methods.’*
>
> Similarly, we have modified certain sentences in the results' discussion to paint a cleaner/more accurate picture. Please refer to the diff file for the complete summary of changes.
>
> >**I am not particularly convinced that settings without HER are very interesting. What is a setting where HER cannot be used?  If that is where VDIFF shines then there should be an evaluation in such a setting directly rather than evaluating in a setting where HER can be used.**
>
> HER is extremely powerful and significantly accelerates performance in sparse reward, goal-based RL. However, there are a few settings where HER cannot be used effectively -
>
> - The goal substitution (using virtual goals) changes the distribution of experience and makes HER biased [1,2]. The negative impact of this bias is visible in 3 environments in which the asymptotic success rate of HER is below VDIFF-R (No HER). We have added the HER plot to Figure 3 to show this difference. If a practitioner only cares about the final success rate, then VDIFF-R (No HER) could be useful in environments where the bias of HER affects performance.
> - HER is an off-policy method and cannot work with on-policy algorithms such as PPO, A3C [4]. VDIFF or VDS have no such constraints as unlike HER, they do not directly modify training data.
> - HER does not work in general contextual RL settings [3]. Goal-conditioned RL is a special case of contextual RL in which the context is a goal. We did not evaluate in the general contextual setting as the main focus of this work was on improving goal-conditioned RL. Future work would aim to apply VDIFF to these general settings.
>
> >**I think it would be good to include the algorithms in the main text if possible. Similarly, I think the selective resampling mentioned in the appendix ought to be mentioned in the main text as it adds complexity.**
>
>
> Thank you for the suggestion, we have moved both the selective resampling section and the VDIFF algorithm pseudocode into the main paper. However, due to space constraints, the selective resampling algorithm pseudocode is still in the appendix.
>
> We hope we have been able to address your concerns in the above clarifications. Kindly let us know if you have additional questions and we would be more than happy to discuss further.
>
> [1]: Lanka, Sameera, and Tianfu Wu. "Archer: Aggressive rewards to counter bias in hindsight experience replay." arXiv preprint arXiv:1809.02070 (2018).
>
> [2]: Schramm, Liam, et al. "USHER: Unbiased Sampling for Hindsight Experience Replay." arXiv preprint arXiv:2207.01115(2022).
>
> [3]: Eimer, Theresa, et al. "Self-paced context evaluation for contextual reinforcement learning." International Conference on Machine Learning. PMLR, 2021.
>
> [4]: Plappert, Matthias, et al. "Multi-goal reinforcement learning: Challenging robotics environments and request for research." arXiv preprint arXiv:1802.09464 (2018).

---

### Official Review · Reviewer_5Ze8 · 2022-10-24

**Confidence:** 4
**Correctness:** 3
**Technical Novelty And Significance:** 2
**Empirical Novelty And Significance:** 2
**Recommendation:** 5

**Clarity, Quality, Novelty And Reproducibility:**

The paper has good clarity overall.

The novelty of the paper dose not seem significant enough, given that the main proposed technique (continuous goal sampling) is extremely simple and seems to have some obvious drawbacks/limitations, and the other automatic curriculum learning algorithm proposed essentially uses a similar idea from existing works.

**Strength And Weaknesses:**

- Strengths:
  - The proposed continuous goal sampling technique and automatic curriculum learning algorithm are very simple and not tied to the base RL algorithms.
  - The paper is well-written and easy to follow overall.
- Weaknesses:
  - In continuous goal sampling, just sampling a new goal every *fixed* number of timesteps seems to have some obvious drawbacks/limitations (and the paper dose not seem to discuss it). It does not really take into account the agent’s current learning progress or the state it is in, but just blindly changing goals when the time meets the pre-designed fixed condition. I could easily imagine some scenarios where this might hurt the sample efficiency, e.g., when the agent is close to reach the current goal and receive the highest reward, the goal (reward function) is suddenly changed. Then what the agent learns before could be wasted.
  - The technical novelty of the paper does not seem significant enough (e.g., sampling goals based on the agent's learning progress has been done before as mentioned in the paper).
  - There seems to be some problems with the hyperparameters settings used in the baseline methods in the experiments. Specifically, in the experiments, the paper mentions "All baseline methods also use SAC with identical hyperparameter settings to enable a fair comparison." I do not understand why the baseline methods have to use the *same* hyperparameter settings as the proposed method to enable a fair comparison (if I understand it correctly). It is common to do separate hyperparameter tuning for different methods when comparing their performance in deep learning.

**Summary Of The Paper:**

This paper aims to accelerate and improve goal-conditioned RL. The main contribution is the continuous goal sampling technique proposed. The idea is very simple. Instead of sampling a goal only at the start of an episode, we can sample a new goal every fixed number of timesteps within an episode. The second contribution is the automatic curriculum learning algorithm proposed, which uses learning progress estimated through value functions to create a self-paced curriculum. These two techniques are both not tied to the base RL algorithms.

**Summary Of The Review:**

This work is well-motivated and the proposed methods are simple and easy to understand. Based on the experimental results, the main proposed technique continuous goal sampling seems to be quite effective in a variety of tasks tested, even though its idea is so simple. However, I'm not sure if the proposed method is compared against baseline methods in a fair way as the authors claimed, making it hard for me to judge how effective the proposed method really is. Also, I think the limitations of continuous goal sampling need to be clearly discussed.

---

> ### Author Response · Authors · 2022-11-14
> **Response to Official Review of Paper3535 by Reviewer 5Ze8**
>
> We thank you for the time and effort you put into this review, it has greatly helped us understand how we can improve our paper.. We provide clarifications and comments for your questions below -
>
> >**In continuous goal sampling, just sampling a new goal every fixed number of timesteps seems to have some obvious drawbacks/limitations.**
>
> Thank you for this keen observation. We had the same intuition and had proposed a strategy to resample selectively i.e., only resample a goal if some criteria is met. More specifically, a goal is resampled only if the expected LP from resampling is greater than the estimated LP of the current goal. However, we found that selective resampling does not have any significant impact on performance. As a result, this subsection was originally placed in the Appendix. We have now moved this into the main paper. While we do not have a concrete answer on why selective resampling does not improve performance, we believe that a possible reason could be the presence of HER, which generates positive reward through virtual goals and thus reduces the importance of receiving actual positive rewards from the environment . It would be interesting to analyze and explore different strategies for resampling, which we leave for future work.
>
> >**The technical novelty does not seem enough.**
>
> We refer you to the global [comment][here] on novelty.
>
>
> >**There seems to be some problems with the hyperparameters settings used in the baseline methods in the experiments. It is unclear why the baseline methods have to use the same hyperparameter settings as the proposed method to enable a fair comparison.**
>
> We would like to offer clarification here:
>
> First, we use an unaltered, off-the-shelf implementation of SAC. More specifically, we use the same hyperparameters and implementation as provided in the popular OpenAI Spinup repository. We find that this off-the-shelf implementation works extremely well in practice, the success rates obtained are generally much better than previous works [1,2], most of which use DDPG. Notably, the performance of even SAC with no curriculum is much better than that of previous works. While it would take more empirical rigor to validate this, we believe that SAC is a better choice than DDPG for the tested benchmark environments.
>
> Second, we tune the individual hyperparameters of the curriculum algorithms. For example, in the case of VDS, we tune the ensemble size and the learning rate (learning rate here refers to the learning rate for the ensemble of value functions, which are separate from SAC).
>
> Thus, the hyperparameters of the curriculum methods have been tuned, but the SAC implementation is fixed and off-the-shelf. The reasoning behind this is that curriculum methods treat the base RL algorithm as a blackbox, and tuning the base RL algorithm for a given curriculum method would go against this. Note that by being off-the-shelf, the SAC implementation has not been tuned to VDIFF (our proposed algorithm) in any way either.
>
> We hope we have been able to address your concerns in the above clarifications. Kindly let us know if you have additional questions and we would be more than happy to discuss further.
>
> [1] : Zhang, Yunzhi, Pieter Abbeel, and Lerrel Pinto. "Automatic curriculum learning through value disagreement." Advances in Neural Information Processing Systems 33 (2020): 7648-7659.
>
> [2] : Fang, Meng, et al. "Curriculum-guided hindsight experience replay." Advances in neural information processing systems 32 (2019).
>
> [here]: https://openreview.net/forum?id=Vk9RH9aL1Yv&noteId=Oo-2o-YHTe

---

> > ### Comment · Reviewer_5Ze8 · 2022-12-05
> > **Update**
> >
> > After some further discussion with other reviewers, I decided to maintain my recommendation to reject. While it's nice the main proposed method is simple yet effective (in some environments), I think the current experimental evidence is not strong enough to demonstrate significant contribution. I believe the paper would benefit from a deeper analysis of the main limitations of the proposed method (helping the reader to better understand when the method works/doe not work and then be more motivated to use it).

---

### Official Review · Reviewer_r5BM · 2022-10-24

**Confidence:** 3
**Correctness:** 3
**Technical Novelty And Significance:** 4
**Empirical Novelty And Significance:** 4
**Recommendation:** 5

**Clarity, Quality, Novelty And Reproducibility:**

The content provided in the paper is mostly clear. The descriptions of the algorithmic ideas are clear. Nevertheless, the presentation of the experiments and associated discussion could be improved (see weaknesses above.)

The algorithmic ideas are simple, but I believe that they are novel.

Given the information provided in the paper, I believe the results should be reproducible.


**Strength And Weaknesses:**

The main strength of this paper is that it presents two clear and testable algorithmic ideas. The ideas are also relatively simple, which would make them easy to implement. Thus, this paper could be of interest to a broad range of RL practitioners.

The main weakness of this paper is the presentation and discussion of the empirical results.

With the first set of experiments, for example, the learning curves for only 9 of the 17 environments are shown. In 6 of the 9 shown, VDIFF outperforms the baselines, but in reality VDIFF outperforms the baseline only 6 of 17 environments (after a close look at the appendix). This disparity is misleading, and is not adequately highlighted. However, it is worth noting that VDIFF is never worse than the baselines (except maybe in 1 case.)

It would be useful to summarize all the data somehow in the main text, instead of including only cherry picked plots.

If possible, it would be useful to increase the number of seeds used in the experiments to at least 10, but 30 would be ideal. More seeds would help to support stronger conclusions.

How is the 95% confidence interval used in the plots computed? Is the distribution of the learning curves assumed to be normal? Is this actually the case?

Minor: The colors used in the plots make the lines hard to distinguish. Consider changing the colors to something with more contrast or even label the lines directly instead of using a legend.

In Figure 1, what is the algorithm labeled Random? It not clear to me from the text.

The paper uses the term “vanilla” throughout to mean “plain”. I don’t think that this term is useful or descriptive and can be confusing. It would be better to simply label each algorithm based on its components, e.g. “SAC with HER”.

In the last paragraph of Section 3.2.1 the paper says that using the smoothed updates provide a cleaner, less noisy LP signal, and that in practice the gain outway the slowdown introduced. Could this be further discussed? Are there experiments that compare the two cases? How big is the difference in practice?


**Summary Of The Paper:**

This paper addresses learning in goal-conditioned RL problems, and presents two contributions. The first is a simple algorithmic technique that continuously resamples goals within an episode, instead of only between episodes, to speed up learning. The second is an automatic curriculum learning method driven by the agent’s value function called VDIFF. VDIFF samples goals for which the agent is demonstrating learning progress. The effectiveness of these techniques, both separately and combined, is supported by empirical experiments in simulated robotic environments.

**Summary Of The Review:**

Overall, the paper presents two clear new algorithmic ideas, but the presentation and discussion of the experiments is lacking. Therefore I would argue to reject.

The paper could be improved by cleaning up the presentation of the experiments and expanding the discussion.

---

> ### Author Response · Authors · 2022-11-14
> **Response to Official Review of Paper3535 by Reviewer r5BM**
>
> We thank you for your insightful review and we are glad that you think this paper could be of interest to a broad range of RL practitioners. We provide clarifications and comments for your questions below -
>
>
> >**With the first set of experiments, for example, the learning curves for only 9 of the 17 environments are shown. This disparity is misleading, and is not adequately highlighted. It would be useful to summarize all the data in the main text.**
>
> Based on your suggestion, we add sentences to explicitly summarize this into the main text. For example, the captions for figures in the paper now include the following sentence -
>
> *‘Note that these plots mainly show the environments on which VDIFF-R improves performance. Complete results can be found in Appendix (linked).’*
>
> Similarly, we add the following summarization sentences in Section 4.2 -
>
> *‘From the results, we first observe that VDIFF-R shows improved sample efficiency or better performance compared to baseline methods in 7 out of the 17 environments. In the other 10 environments, VDIFF-R matches the performance of baseline methods.’*
>
> These are just 2 examples of sentences added to make the discussion of the results clearer. We add summarising sentences to other discussion sections as well. Please refer to the diff file for the complete summary of changes.
>
>
> >**If possible, it would be useful to increase the number of seeds used in the experiments to at least 10, but 30 would be ideal. More seeds would help to support stronger conclusions.**
>
> We are currently in the process of running all experiments with a larger number of seeds. However, since we are in an academic setting with limited compute, it will take some time to obtain the full set of results. We will add these results into the paper as we obtain them.
>
>
> >**How is the 95% confidence interval used in the plots computed? Is the distribution of the learning curves assumed to be normal? Is this actually the case?**
>
> The 95% confidence interval is over the mean of the success rates over 6 seeds. Yes, the learning curves are assumed to be normal. However, some environments have higher variance and more seeds will help in getting a better estimate. We are in the process of doing this and will be updating the plots as soon as possible.
>
>
> >**The colors used in the plots make the lines hard to distinguish.**
>
> Thank you for your suggestion, we have changed the colors of lines on certain figures to make them clearer.
>
> >**In Figure 1, what is the algorithm labeled Random? The paper uses the term “vanilla” throughout to mean “plain”. It would be better to simply label each algorithm based on its components, e.g. “SAC with HER”.**
>
> We apologize for the lack of clarity. Random refers to SAC (with NO HER). We have added a sentence to describe Random in the main text. We have also modified the algorithm labels to increase clarity.
>
>
> >**The paper says that using the smoothed updates provide a cleaner, less noisy LP signal, and that in practice the gain outway the slowdown introduced. Could this be further discussed?**
>
> We present a new ablation study (Section 4.6 of the paper) to empirically validate and discuss the claim that target value networks lead to smoother LP signals. The full result plots for this study have also been added to the paper (in the Appendix).
>
> We hope we have been able to address your concerns in the above clarifications. Kindly let us know if you have additional questions and we would be more than happy to discuss further.

---

> > ### Comment · Reviewer_r5BM · 2022-11-16
> > **Reply**
> >
> > Thanks for answering my questions, and for the clarifications.
> >
> > Are you able to describe exactly how you computed the 95% confidence intervals for your plots (e.g. with a formula)? I want to make sure I am able to interpret the plots correctly.

---

> > > ### Author Response · Authors · 2022-11-17
> > > **Response to Reviewer r5BM's question on confidence intervals**
> > >
> > > > **Are you able to describe exactly how you computed the 95% confidence intervals for your plots (e.g. with a formula)? I want to make sure I am able to interpret the plots correctly**.
> > >
> > >
> > > Thank you for following up, we appreciate the time you took to review our rebuttal. We follow the implementation used by the seaborn library to compute a 95% CI. Seaborn uses bootstrap confidence intervals which are described below and at this [link][here].
> > >
> > > At the end of every training epoch, we have N calculated success rates, where N is the number of seeds (in our case, 6). Let M (typically >=1000) be the number of bootstrap iterations. Then the confidence interval for that epoch is computed as follows:
> > >
> > > 1. U = [ ]
> > > 2. Draw N samples with replacement from the N data points.
> > > 3. Compute the mean u* of the N samples. Add u* to U.
> > > 4. Repeat step 2 and 3, M times.
> > > 5. Sort U
> > > 6. The confidence bounds are obtained using the middle 95% of U. For example, If M=1000, then the confidence bounds for the interval would be the 25th and 975th element.
> > >
> > > We use an epoch length of 20000 training timesteps i.e., 20000 steps/interactions with the environment. Thus, for every 1 million training timesteps, we have 50 epochs and at the end of each epoch, success rate for an agent is obtained by running the agent on 200 test episodes with randomly sampled goals.
> > >
> > > Finally, note that in curriculum learning, it is necessary to run separate testing episodes as we do above. This is because the goal distribution during training is chosen by the curriculum method itself, and reporting training episode returns would be a biased metric.
> > >
> > > We hope we have been able to clarify your doubts through this answer. Kindly let us know if you have additional doubts or require further clarification.
> > >
> > > [here]:https://elizavetalebedeva.com/bootstrapping-confidence-intervals-the-basics/

---

> ### Comment · Reviewer_r5BM · 2022-12-05
> **Update**
>
> After some further discussion with the other reviewers, I believe the experiments and discussion are still lacking. I think what is missing is a focussed example to help the reader carefully build intuition about the new method. The existing experiments on a large variety of tasks are good, but they are too broad and numerous for a reader to understand them intricately; a smaller and simpler example that could be thoroughly understood would help greatly. This example could help highlight the advantages of the proposed method, and help the reader understand why, and when it might be useful. This is present to some extent in the existing experiments and analysis, but I think an illustrative example would help increase the impact for a reader.
>
> Therefore, I maintain my recommendation to reject. However, I strongly encourage the authors to refine this work, and resubmit some time in the future.

---

### Author Response · Authors · 2022-11-14
**Global Comments and Revision Summary in response to reviews of Paper 3535**

We thank the reviewers for their time and effort in reviewing this paper. The insightful comments received have helped us improve this paper. In addition to the individual review responses, we provide a summary of revisions and general comments below.


**Ablation on target value networks for computing learning progress (LP)** : In this work, we proposed to use the target value function for LP computation. We claimed that using the target value function provides a cleaner LP signal. We present additional results to empirically validate this claim by comparing target value functions against standard (no polyak averaging) value functions for computing LP. The latter is very similar to the definition of LP used by SPaCE [1], a prior work which also uses value functions for LP computation. Through these results, we show that target value functions lead to a cleaner LP signal and can also have a significant impact on performance.


**Novelty**: We understand the reviewers’ concerns on the novelty of VDIFF. The main purpose of this work was to introduce two simple ideas that an RL practitioner can easily integrate into their goal-based RL framework. While learning progress (LP) has been commonly used in prior works, it has generally either been applicable only to discrete task-based domains [1,2] or has required the use of additional learning components such as Gaussian Mixture Models (GMM) [3] for continuous task domains.
To the best of our knowledge, we are the only work to use LP in continuous goal/task domains without introducing any additional learning component. We believe that compared to prior works in the area, VDIFF is significantly simpler to integrate into existing goal-conditioned RL frameworks.

Similar to our method, SPaCE [1] also uses value functions for LP computation. However, SPaCE was proposed for contextual RL in discrete settings and thus has a fundamentally different scope from ours (also mentioned in the original SPaCE paper).
Additionally, through new results (bullet point above), we show that using the target value network can significantly improve performance compared to standard value functions (which are used by SPaCE).


**Selective Resampling of Goals**: VDIFF samples a new goal only if the expected LP of sampling a new goal is greater than the estimated LP of the current goal. This implementational detail, which we formally refer to as selective resampling, was originally placed in the appendix as it did not seem to have an impact on performance, particularly in the HER domain. However, as suggested by reviewer aTaV, we have moved this into the main paper to bring it into focus. We have also moved Algorithm 1 into the main text based on reviewer aTaV‘s suggestions.


**Unclear/misleading writing**: We thank reviewers for pointing this out. To remedy this, we try to improve clarity of writing in certain sections of the paper. In particular, based on reviewer r5BM’s suggestions, we highlight that the results shown in the main paper mainly show the environments on which VDIFF-R improves performance, and that the complete benchmark results can be found in the Appendix. We also add sentences to better summarize the results in the main paper. Similarly, based on reviewer aTaV’s suggestions, we more concretely clarify that while resampling generally helps, VDIFF-R matches but does not improve performance over HER-R. The abstract has been modified to reflect the same.


[1] : Eimer, Theresa, et al. "Self-paced context evaluation for contextual reinforcement learning." International Conference on Machine Learning. PMLR, 2021.

[2] : Matiisen, Tambet, et al. "Teacher–student curriculum learning." IEEE transactions on neural networks and learning systems 31.9 (2019): 3732-3740.

[3] : Portelas, Rémy, et al. "Teacher algorithms for curriculum learning of deep rl in continuously parameterized environments." Conference on Robot Learning. PMLR, 2020.

---

### Decision · Program_Chairs · 2023-01-20

**Decision:**

Reject

**Justification For Why Not Higher Score:**

This paper was on the borderline. The reviewers appreciated the simplicity of the work and the two ideas. However, after a discussion with the reviewers, there was a consensus that a better experimental design and qualitative analysis was required for this work to be a clear contribution to the research community. The reviewers felt that a better explanation (via experiments and analysis) of how the two ideas combine would have made the paper acceptable (and that would be a good direction for a future revision). Additional effort on the experiment design would also make the resulting analysis more convincing.

**Justification For Why Not Lower Score:**

N/A

**Metareview: Summary, Strengths And Weaknesses:**

Summary:
This paper presented two contributions for goal conditioned episodic RL problems.  The first is sampling goals continuously throughout the episode (instead of at the end in the manner of Hindsight Experience Replay).  The other is an automated curriculum method which is informed by learning progress across time (VDIFF).  The method shows improvements on some problems in the experiments.

Strengths:
The core contribution is presented clearly, the problem setting is well motivated, and the results look promising to the reviewers.

Weaknesses:
The reviewers found the evidence in the paper to be lacking, and the mixture of the two contributions were muddled.  The revisions to the paper by the authors did partially rectify the reviewer concerns, but not fully.

Additional experimental limitations were noted by a separate examination of the paper, after the AC-reviewer meeting.
- "All baseline methods also use SAC with identical hyperparameter settings to enable a fair comparison." This is not fair, it is misleading.
- Is the initial exploration period of SAC included, as in the SAC paper?  This is crucial for good performance.
- Standard normal CI must be justified in the small sample regime (6 seeds). Student-t CI would be much better.
- The original HER paper used DDPG, its problematic this paper compares against SAC+HER, same issue for VDS
- A better configuration of SAC is well known and widely cited (https://arxiv.org/abs/1812.05905)
- It is unclear if all differences can be explained by untuned baselines. These changes from DDPG to SAC are confusing.

**Summary Of Ac-Reviewer Meeting:**

A meeting of three of the reviewers was held (two weakly positive, one weakly negative).  The missing reviewer (aTaV was not available) was weakly negative on the paper.

There was agreement on the main paper contributions as described by reviewer UKQZ.

1. the paper presents two, independent ideas (so that's bad compared to a one strong idea)
2. the ideas are simple yet novel (ie. these concrete ideas aren't likely part of previous work, but it's possible a practitioner may try them whether reading this work or not)
3. the results are promising (good)

The discussion among the reviewers led to the experiments were flawed for conveying the main points of the paper.  The reviewers at the meeting were in consensus that an improved qualitative analysis of the experimental results would make the paper much stronger, and the paper did not adequately explain how the two ideas build on each other.